# Genetic and environmental drivers of large-scale epigenetic variation in *Thlaspi arvense*

**Dario Galanti**[1], **Daniela Ramos-Cruz**[2,3], **Adam Nunn**[4,5], **Isaac Rodríguez-Arévalo**[2,3], **J. F. Scheepens**[6], **Claude Becker**[2,3], **Oliver Bossdorf**[1]*

1 Plant Evolutionary Ecology, Institute of Evolution and Ecology, University of Tübingen, Tübingen, Germany, 2 Gregor Mendel Institute of Molecular Plant Biology, Austrian Academy of Sciences, Vienna BioCenter (VBC), Vienna, Austria, 3 LMU Biocenter, Faculty of Biology, Ludwig Maximilians University Munich, 82152 Martinsried, Germany, 4 ecSeq Bioinformatics GmbH, Leipzig, Germany, 5 Institute for Computer Science, University of Leipzig, Leipzig, Germany, 6 Plant Evolutionary Ecology, Institute for Ecology, Evolution and Diversity, Faculty of Biological Sciences, Goethe University Frankfurt, Frankfurt am Main, Germany

* oliver.bossdorf@uni-tuebingen.de

## Abstract

Natural plant populations often harbour substantial heritable variation in DNA methylation. However, a thorough understanding of the genetic and environmental drivers of this epigenetic variation requires large-scale and high-resolution data, which currently exist only for a few model species. Here, we studied 207 lines of the annual weed *Thlaspi arvense* (field pennycress), collected across a large latitudinal gradient in Europe and propagated in a common environment. By screening for variation in DNA sequence and DNA methylation using whole-genome (bisulfite) sequencing, we found significant epigenetic population structure across Europe. Average levels of DNA methylation were strongly context-dependent, with highest DNA methylation in CG context, particularly in transposable elements and in intergenic regions. Residual DNA methylation variation within all contexts was associated with genetic variants, which often co-localized with annotated methylation machinery genes but also with new candidates. Variation in DNA methylation was also significantly associated with climate of origin, with methylation levels being lower in colder regions and in more variable climates. Finally, we used variance decomposition to assess genetic versus environmental associations with differentially methylated regions (DMRs). We found that while genetic variation was generally the strongest predictor of DMRs, the strength of environmental associations increased from CG to CHG and CHH, with climate-of-origin as the strongest predictor in about one third of the CHH DMRs. In summary, our data show that natural epigenetic variation in *Thlaspi arvense* is significantly associated with both DNA sequence and environment of origin, and that the relative importance of the two factors strongly depends on the sequence context of DNA methylation. *T. arvense* is an emerging biofuel and winter cover crop; our results may hence be relevant for breeding efforts and agricultural practices in the context of rapidly changing environmental conditions.

**Data Availability Statement:** Seed material from the sequenced lines was collected complying with the Nagoya protocol (permit number: TREL1734890A/19) and is available at the

Nottingham Arabidopsis Stock Centre (NASC) under stock numbers N950001 to 950204. Genomic and bisulfite sequencing raw data are available on the ENA Sequence Read Archive (www.ebi.ac.uk/ena/) under study accession number PRJEB50950. Reference genome and annotations were previously published by Nunn et al. (2022). From the annotation, we subset confident de novo gene annotations with an annotation edit distance score < 1.0 (source: T_arvense_v2). Mean and weighted methylation values extracted from sequencing data, DMRs, Gene Body Methylation classification and GWA results (-log(p)>1) in a format compatible with the Integrative Genomics Viewer (software. broadinstitute.org/software/igv/) are available on Zenodo (doi.org/10.5281/zenodo.6361977). All the code used in this study is available and documented on Github. Pipelines for methylation and DMR calling from WGBS data are on the EpiDiverse Github (https://github.com/EpiDiverse). The workflow for downstream analysis of methylation data is on https://github.com/Dario-Galanti/WGBS_downstream. Scripts for downstream processing of DMRs and DMRs variance decomposition are on https://github.com/Dario-Galanti/popDMRs_refine_VCA. Scripts for variant calling, filtering and imputation, performed on the Baden-Württemberg BinAC cluster, are on https://github.com/Dario-Galanti/BinAC_varcalling. Finally, scripts for running GWA analysis and the enrichment of a priori candidates are on https://github.com/Dario-Galanti/multipheno_GWAS.

**Funding:** This work is part of the European Training Network EpiDiverse (https://epidiverse.eu), which received funding from the EU Horizon 2020 program under Marie Skłodowska-Curie grant agreement No 764965. This work was also supported by the European Union's Horizon 2020 research and innovation program via the European Research Council (ERC) Grant agreement No. 716823 "FEAR-SAP" to C.B., by the Austrian Academy of Sciences and by the German Research Foundation (DFG) through grant no INST 37/935-1 FUGG. The funders had no role in study design, data collection and analysis, decision to publish, or preparation of the manuscript.

**Competing interests:** The authors have declared that no competing interests exist.

## Author summary

Variation within species is an important level of biodiversity, and it is key for future adaptation. Besides variation in DNA sequence, plants also harbour heritable variation in DNA methylation, and we want to understand the evolutionary significance of this epigenetic variation, in particular how much of it is under genetic control, and how much is associated with the environment. We addressed these questions in a high-resolution molecular analysis of 207 lines of the common plant field pennycress (*Thlaspi arvense*), which we collected across Europe, propagated under standardized conditions, and sequenced for their genetic and epigenetic variation. We found large geographic variation in DNA methylation, associated with both DNA sequence and climate of origin. Genetic variation was generally the stronger predictor of DNA methylation variation, but the strength of environmental association varied between different sequence contexts. Climate-of-origin was the strongest predictor in about one third of the differentially methylated regions in the CHH context, which suggests that epigenetic variation may play a role in the short-term climate adaptation of pennycress. As pennycress is currently being domesticated as a new biofuel and winter cover crop, our results may be relevant also for agriculture, particularly in changing environments.

## Introduction

Besides variation in DNA sequence, natural plant populations usually also harbour variation in epigenetic modifications of the DNA. This is particularly well documented for DNA methylation, usually referring to the addition of a methyl group to the 5$^{th}$ atom of the cytosine ring, a modification associated with silencing of transposable elements (TEs) and the regulation of gene expression. Variation in DNA methylation can arise if methylation marks are altered by chance during mitosis or meiosis (epimutations) [1,2], or if they are induced in response to environmental changes [3,4]. Some DNA methylation differences are stably inherited through meiosis, which has led some to hypothesize that DNA methylation variation could be under natural selection and contribute to adaptation [5–7]. These ideas are fuelled by the observation that DNA methylation variation in natural plant populations is often non-random and geographically structured [8–12]. However, the DNA methylation variation observed in the field is always a combination of stable (= heritable) and plastic (= non-heritable) components. In order to tease these apart and describe the heritable component of DNA methylation variation, one must analyse the offspring of different populations grown in a common environment. To date, common-environment analyses of natural DNA methylation variation that cover many populations and broad environmental gradients are still rare.

In plants, DNA methylation can occur in the three sequence contexts: CG, CHG and CHH (where H is A, T or C). Distinguishing between these contexts is sensible because they differ in the molecular machineries for depositing, maintaining and removing methylation [13,14], which has consequences for their dynamics and stability. In *Arabidopsis thaliana*, CG methylation (mCG) is mostly maintained in a copy-paste manner during replication, CHG methylation (mCHG) by DNA-histone methylation self-reinforcing loops and CHH methylation (mCHH) by recursive *de-novo* methylation deposited by the RNA-directed DNA methylation pathway (RdDM) and partially by CMT2 [13,14]. In addition, CHG and CHH methylation partially share maintenance pathways [15,16]. Overall, there is a gradient of similarity and decreasing stability from CG to CHG to CHH. Although less stable, CHH is the most abundant context and often the most responsive to stresses [17]. Besides the sequence contexts, the

dynamics of DNA methylation also strongly depend on the genomic features in which it occurs. While heterochromatic regions and TEs are usually heavily methylated to repress transcription, methylation is often lower and more variable in genes and regulatory regions [18–20]. In addition, while DNA methylation is almost exclusively a repressive mark on TEs and in regulatory regions, its function is less clear in gene bodies, as several constitutively expressed housekeeping genes often harbour CG but not CHG and CHH methylation in their coding sequences (CDS) [20,21]. If methylation in different genomic features has different functions, then also different selective pressures are to be expected [22]. Finally, for both influences of sequence context and genomic features on methylation variation, there appears to be high species-specificity in plants [20].

To study such complex dynamics, DNA methylation can be quantified at different levels, from global (or genome-wide) methylation, to average methylation limited to sequence contexts or genomic features, to the methylation of genomic regions or individual positions. While genetic single nucleotide polymorphisms (SNPs) can have large effects, this does not seem to be the case for DNA methylation polymorphisms, which affect transcription only when accumulating over a broader genomic region [23–25]. For this reason, the study of differentially methylated regions (DMRs) became very popular in high-resolution studies [11,12,18,26].

Given the complex molecular machinery for regulation and maintenance of DNA methylation, it is not surprising that previous studies have demonstrated various kinds of genetic control over DNA methylation variation. Genetic polymorphisms can control DNA methylation in *cis*, for example, when a TE insertion next to a gene promoter induces the methylation of the latter [25], or in *trans*, when genetic mutations affect genes involved in the DNA methylation machinery [11,12,27]. In the latter case, variation in individual DNA loci often affects methylation levels across the entire genome. In addition, a number of genes have been found to affect methylation levels indirectly, acting upstream or in aid of the methylation machinery. In particular, ubiquitination, a post-translational modification affecting histone tails and protein turnover, affects DNA methylation in plants and animals in several ways [28–33]. For example, in plants ORTH/VIM E3 ubiquitin ligases recruit DNA METHYLTRANSFERASE 1 (MET1) for methylation maintenance through ubiquitination of histone tails [30,31]. However, in spite of this functional understanding of several mechanisms of genetic control, we still lack a good understanding of the degree of genetic determination of DNA methylation variation in wild plant populations.

If DNA methylation variation is under natural selection–whether independently from DNA sequence or linked to it–we expect this to result in patterns of association between methylation variation and the environment. Several previous studies indeed found correlations between methylation patterns and habitat or climate in different plant species [8–12,34]. However, most of these studies were either conducted in the field, based on only few natural populations, or used low-resolution molecular methods, which limited their generalizability and/or their power to detect environment-methylation associations and to separate genetically controlled from independent components of DNA methylation variation [5]. The only available data that does not suffer from any of these limitations comes from *Arabidopsis thaliana* [11,12,18], a plant with an exceptionally small and simple genome, with low numbers of TEs, and low global DNA methylation [35]. Given these unrepresentative genomic properties, it is currently unclear to which extent findings from population epigenomic studies with *A. thaliana* can be generalised across the plant kingdom. As the abundance and genomic distribution of TEs is a major driver of variation in DNA methylation, species with higher TE contents could differ not only in the extent of DNA methylation, but also in the dynamics of epimutation accumulation, and the DNA methylation-based machinery controlling TE mobility. To

understand the extent of these differences, and the genetic and environmental drivers of natural DNA methylation variation, it is critical to expand our scope and collect large-scale, high-resolution data also for other plant species.

Here, we present a detailed genomic analysis of 207 lines of the plant *Thlaspi arvense* (field pennycress) that we collected across a latitudinal gradient in Europe, cultivated in a common environment, and profiled for genomic and epigenomic variation. Like *A. thaliana*, *T. arvense* is an annual and mostly selfing member of the Brassicaceae family, but it has a significantly larger genome of approx. 500Mb, which is richer in TEs and DNA methylation [36]. The species is an interesting study object also because it is currently being domesticated into a new biofuel and cover crop [37–41]. The genomic work with *T. arvense* is facilitated by recently published high-quality reference genomes [36,42]. In our study, we demonstrate that European populations of *T. arvense* harbour substantial natural epigenetic variation, which is associated with DNA sequence variation as well as with climate of origin, but in a highly context-dependent manner. In our data, genetic variation was generally the stronger predictor of DNA methylation variation. Genome-wide association analyses identified several candidate loci, but there was a fraction of the DNA methylation variation that was most strongly associated with climate of origin, suggesting a link with climate adaptation.

## Results

The 207 *Thlaspi arvense* lines we worked with came from 36 natural populations which we sampled across Europe in 2018, on a latitudinal gradient from Southern France to Central Sweden, with three populations each in Southern France and The Netherlands, seven in Southern Germany, eight in Central Germany and South Sweden, respectively, and another seven populations in Central Sweden (Fig 1A and S1 Table). In each population, we collected seeds of 4–6 different lines (S1 Table). We grew all lines under common environmental conditions, extracted their DNA and generated Whole Genome Sequencing (WGS) and Whole Genome Bisulfite Sequencing (WGBS) libraries, which upon deduplication, were sequenced with an average coverage of 19.7x and 30.3x, respectively (S2 Table). Bisulfite non-conversion rates were calculated from chloroplast DNA and ranged between 0.14 and 1.9% (S2 Table). Variant calling retrieved around nine million SNPs and short INDELs with genotypes called in >90% of the lines. Methylation calling retrieved about 16 million, 18.4 million and 95.3 million positions in CG, CHG and CHH contexts respectively, with up to 25% missing calls per position. The global weighted DNA methylation, calculated as the ratio between all methylated and all total read counts at every analysed cytosine [43], was estimated at 16.9% (average of all lines).

We found significant genetic and epigenetic population structure across Europe. A principal component analysis (PCA) based on genetic variants showed two main clades: a larger one including almost all lines from France, Germany and the Netherlands, and a smaller one that consisted almost exclusively of Swedish lines (Fig 1B). The larger clade also showed a clear latitudinal gradient. PCAs based on DNA methylation variation generally also found two major clades, with the CG methylation-based patterns most closely resembling the genetically-based ones, and a decreasing similarity between genetic and epigenetic population structure from CG to CHG to CHH methylation (Fig 1B and S1). Restricting methylation to specific genomic features also revealed that mCG of genes and promoters has stronger geographic patterns that methylation of TEs (S1 Fig).

### Average methylation

To understand the structure of DNA methylation variation in *T. arvense*, we first examined patterns of weighted average methylation [43] across all lines. We not only distinguished

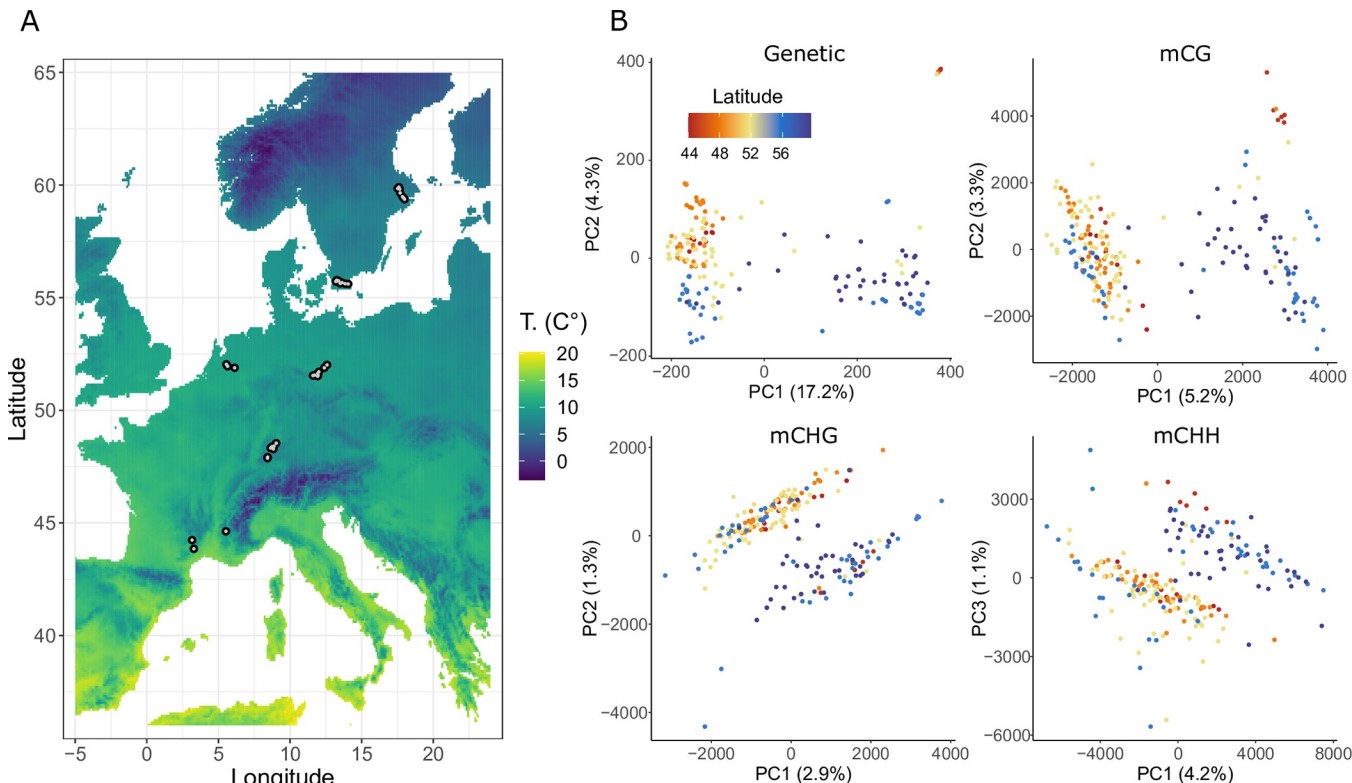

**Fig 1. Geographic distribution and population structure of the 207 sampled *Thlaspi arvense* lines.** (A) Geographic locations of the 36 populations. The background colours are gridded satellite data of average daily temperature (T.) from the Copernicus programme [44]. (B) PCA plots of all 207 lines based on DNA sequence ("Genetic") and DNA methylation in different sequence contexts ("mCG", "mCHG" and "mCHH").

between the three sequence contexts CG, CHG and CHH, but we also assigned cytosines to different genomic features: CDS, introns, promoters, TEs and intergenic regions. For genes and TEs, we used available annotations [36], while for promoters we considered the 2 kb upstream sequences of genes (or until the boundary of the previous gene if closer). We considered intergenic space, anything not belonging to these categories. Across all genomic features, the average methylation was much higher in CG context than in CHG and CHH; for the latter two it was generally similar (Fig 2A). TEs were the most highly methylated genomic features, followed by intergenic regions, whereas promoters and especially gene bodies (CDS and introns) showed very low average methylation (Fig 2A). For instance, while for CG sites in TEs the weighted average methylation was around 80%, it was below 2% for CHH sites in genes. Although these patterns are conserved in the whole collection, there is large residual variation between lines, which is particularly high in TEs (up to 12%) and decreases gradually moving to intergenic regions, promoters and particularly genes (Fig 2A). Finally, partially due to TEs covering about 60% of the *T. arvense* genome [36], its global weighted methylation of 16.9% (average of all lines) is much higher than that of *A. thaliana* (5.8%)[12] and many other Brassicaceae [20].

To better understand the observed values of average methylation, and in particular the low gene body methylation, we further examined the distributions of methylation values of individual CDS, TEs and promoters, averaging across all lines. Interestingly, while context-specific methylation levels were very consistent for TEs, almost exclusively methylated, we found

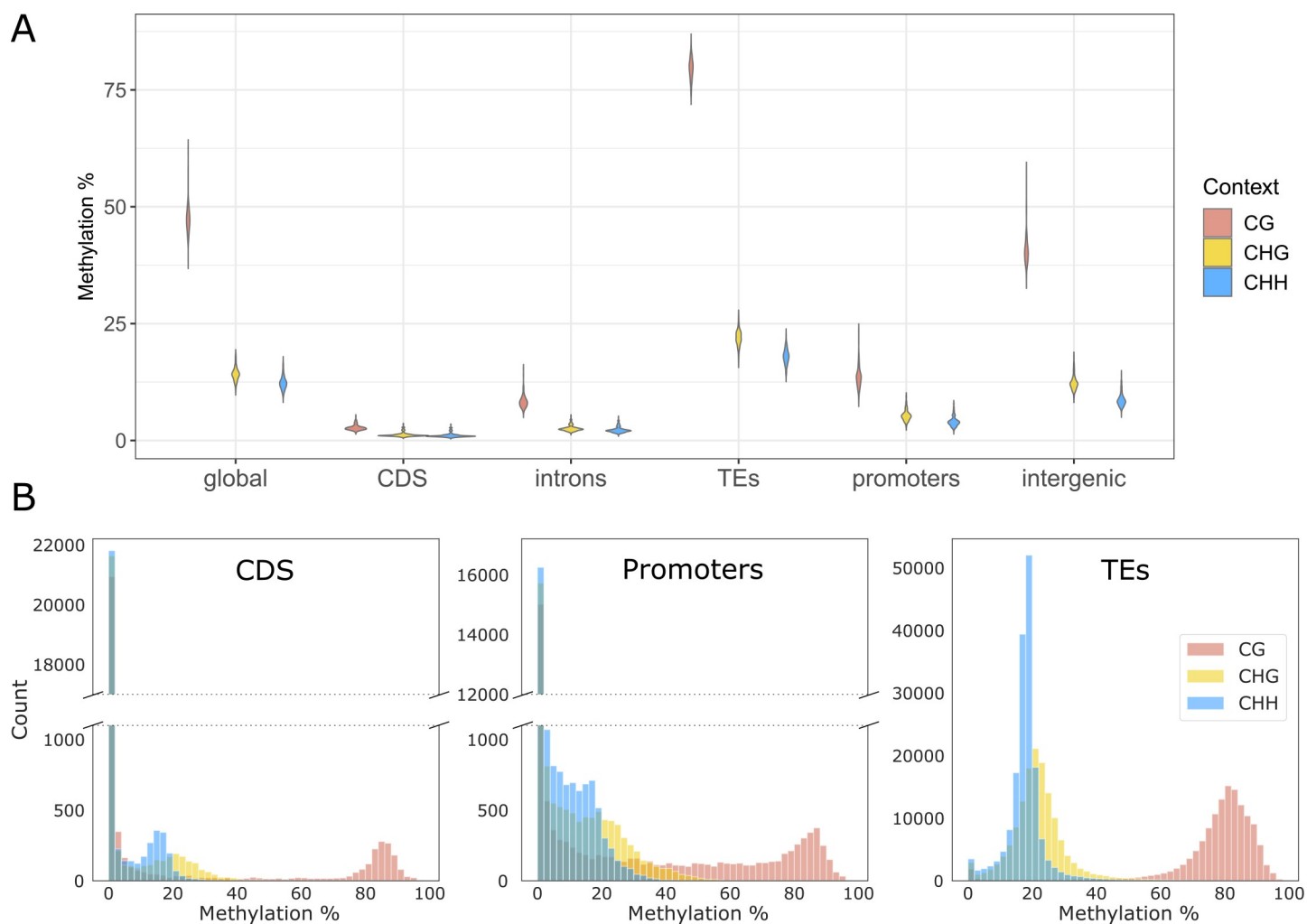

**Fig 2. Average methylation and distributions of methylation values for different sequence contexts and genomic features in *T. arvense*.** (A) Weighted average methylation levels of genomic features; violin plots represent variation between lines. (B) Distributions of individual methylation values for coding sequences (CDS), promoters and transposable elements (TEs) obtained averaging across all 207 lines.

bimodal distributions for CDS and promoters, with a large majority of unmethylated and a smaller fraction of methylated features (Fig 2B). Using a binomial test [20,45], we found than only a small portion of genes is significantly methylated in each sequence context (7.5, 6.5 and 7.3% on average for CG, CHG and CHH respectively), with rather small variation between lines (S3 Table). Intersecting genes consistently methylated (methylated in at least 70% of the lines) in each of the three sequence contexts, we confirmed that a large fraction of these was methylated in all context, showing a TE-like methylation signature (TEm), and a much smaller fraction was methylated only in CG, showing a gene body methylation signature (gbM) (S2A Fig)[36]. Moreover, the fraction of methylated genes, tended to cooccur with TEs, since TEm genes were about eight times more likely than the average gene to overlap with TEs, and gbM genes were twice as likely. Even though many TEm genes might be pseudogenes, a gene ontology (GO) enrichment analysis found enrichment for some housekeeping-like GO terms such as nucleotide biosynthesis and protein catalysis (S2B Fig). In contrast, the few genes methylated only in CG, were only enriched for few molecular functions (S2B Fig).

## Genetic basis of methylation variation

To understand the genetic basis of the observed methylation variation, we employed genome-wide association (GWA) analyses that tested for statistical associations between every biallelic genetic variant and the average methylation of every sequence context and genomic feature (S4 Table). For this analysis we used the (unweighted) mean methylation, as weighted methylation is strongly influenced by structural and copy number variants, which could distort GWA and produce misleading results when looking for individual genes affecting methylation levels. We restricted our analyses to genetic variants with a minor allele frequency (MAF) $\geq 0.04$; however, repeating all analyses with a MAF>0.01 did not influence the results relevantly. Since large numbers of unmethylated genes (Fig 2B) could potentially obscure association patterns in methylated genes, we re-ran these analyses for average methylation levels based only on genes with methylation > 5% (across all lines). In all GWA analyses, we corrected for population structure using an Isolation-By-State (IBS) distance matrix. Although our experimental design and number of lines hardly provided sufficient power to meet a full Bonferroni threshold, we found that many of the genetic variants that were most strongly associated with methylation levels were close to genes with predicted functions related to DNA methylation (Figs 3A, 3D and S3 Fig). For instance, one strong candidate was an orthologue of *ARGONAUTE 9 (AGO9)*, coding a DICER-like protein involved in RNA silencing; *AGO9* natural variation is associated with mCHH in TEs in *A. thaliana* [12]. Another candidate was an orthologue of *DOMAINS REARRANGED METHYLTRANSFERASE 3 (DRM3)*, which despite being catalytically mutated, is necessary for RdDM and non-CG methylation maintenance in Arabidopsis [46–48]. Reflecting the multigenic basis of methylation, even the higher -log(p) variants had relatively small size effects of about 1.5% methylation (Fig 3C).

To confirm the suspected enrichment of methylation-related genes among stronger associations, we conducted an enrichment analysis based on all genetic variants within 20kb from *a priori* candidate genes–orthologues of *A. thaliana* genes known to affect methylation (S5 Table). For many genomic features and sequence contexts, we indeed found an enrichment of these *a priori* candidates among the genetic variants most strongly associated with average methylation levels (e.g. mCG in Fig 3B), but in most cases the top variants were not neighbouring any *a priori* candidates (drop of the enrichment for high -log(p) thresholds in mCHG and mCHH in Fig 3B; see S3 Fig for more results). Nevertheless, a search of the neighbouring regions of these variants identified several new candidates that may not affect methylation directly, but have predicted functions with a potential for indirect effects on DNA methylation. These include e.g. the histone deacetylase *SIRTUIN 1 (SRT1)*, the DNA-damage-repair/toleration *(DRT111), the DNA-repair gene STRUCTURAL MAINTENANCE OF CHROMOSOMES 5 (SMC5)* and several E3 ubiquitin ligases such as F-box transcription factors and RING-H2 finger proteins (Fig 3; see S6 Table for all genes located within 15kb from variants significant at -log(p) > 5). Overall, our results showed that natural DNA methylation variation in *T. arvense* was significantly associated with underlying DNA sequence variation, but only some of the top genetic variants were known methylation machinery genes, whereas there were many additional, less well-characterized genes that appeared to play a role, possibly through less direct effects on methylation.

The GWA results strongly differed between sequence contexts, with a unique profile of genetic variants associated with average mCG, while the results were very similar for mCHG and mCHH (Figs 3A, 3D and S3 Fig). In mCG, some of the top candidates were *AGO9*, the methyltransferase *DRM3*, the F-box/WD-40 repeat-containing gene *Tarvense_02099*, involved in histone methylation, and two orthologues of the SWI/SNF chromatin remodelling component *BAF60* (S6 Table). In mCHG and mCHH, the strongest associations included *SRT1*,

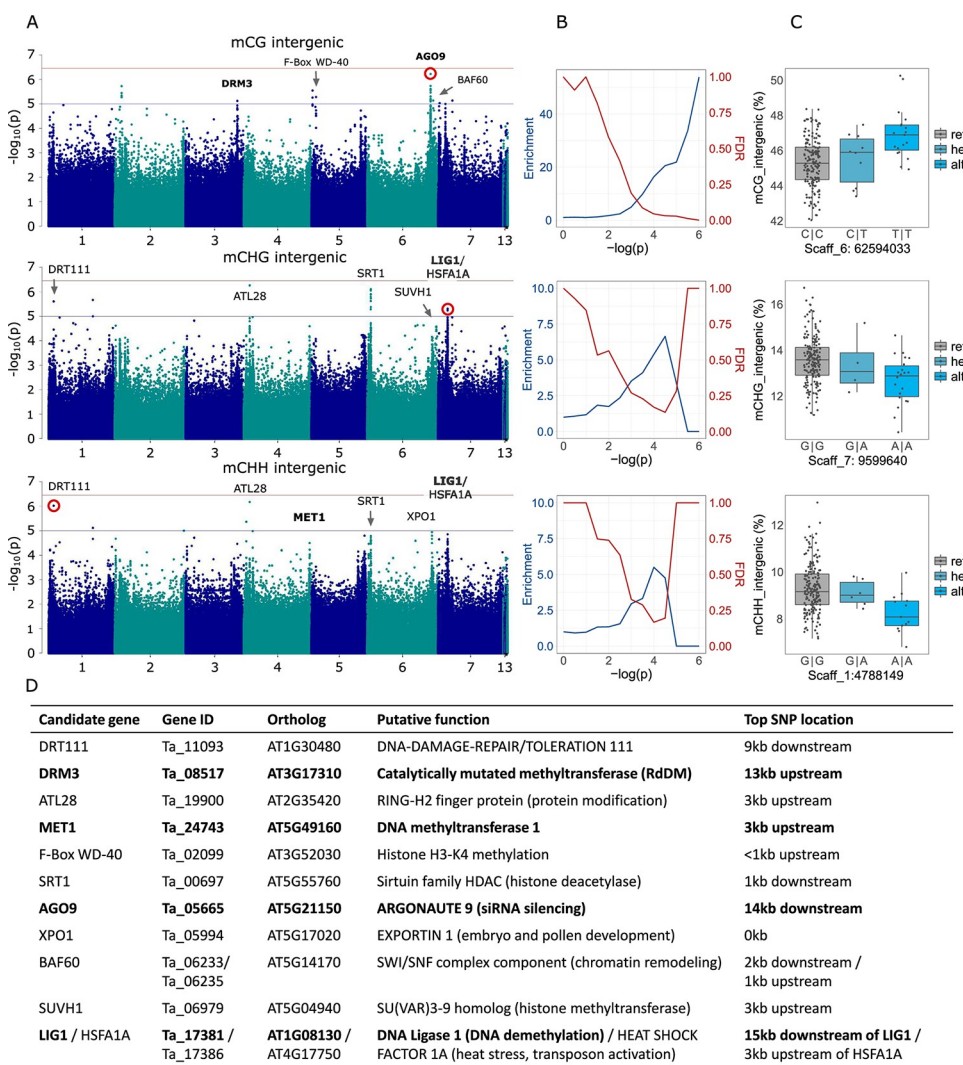

**Fig 3. Genome-wide association analyses for genetic control of average DNA methylation.** We show only the results for intergenic methylation; for full results see S3 Fig. (A) Manhattan plots, with the top variants labelled with the neighbouring genes potentially affecting methylation. The genome-wide significance (horizontal red lines), was calculated based on unlinked variants as in Sobota et al. (2015) [49], the suggestive-line (blue) corresponds to–log(p) = 5. (B) Corresponding to each Manhattan plot on the left, enrichment of *a priori* candidates and expected false discovery rates (both as in Atwell et al. 2010 [50]) for stepwise significance thresholds. (C) The allelic effects of the red-marked variants in the corresponding Manhattan plots on the left, with genotypes on the x-axes and the average methylation on the y-axes. (D) The candidate genes marked in panel A, their putative functions and distances to the top variant of the neighbouring peaks. Bold font indicates *a priori* candidates that were included in the enrichment analyses.

*SMC5*, the *DNA LIGASE 1 (LIG1)*, involved in DNA demethylation, and *DRT111*. Lastly, we tested whether variation in number of gbM genes between lines was associated to genetic variants and detected a clear peak in Scaffold_3, including, among a few additional genes, *LOG2--LIKE UBIQUITIN LIGASE3 (LUL3)*, which codes a ubiquitin ligase (S2C Fig).

## Methylation relationships with climate of origin

To test for environmental associations of methylation variation, we compiled bioclimatic data (see Methods section for details) for our 36 study populations and analysed the relationships

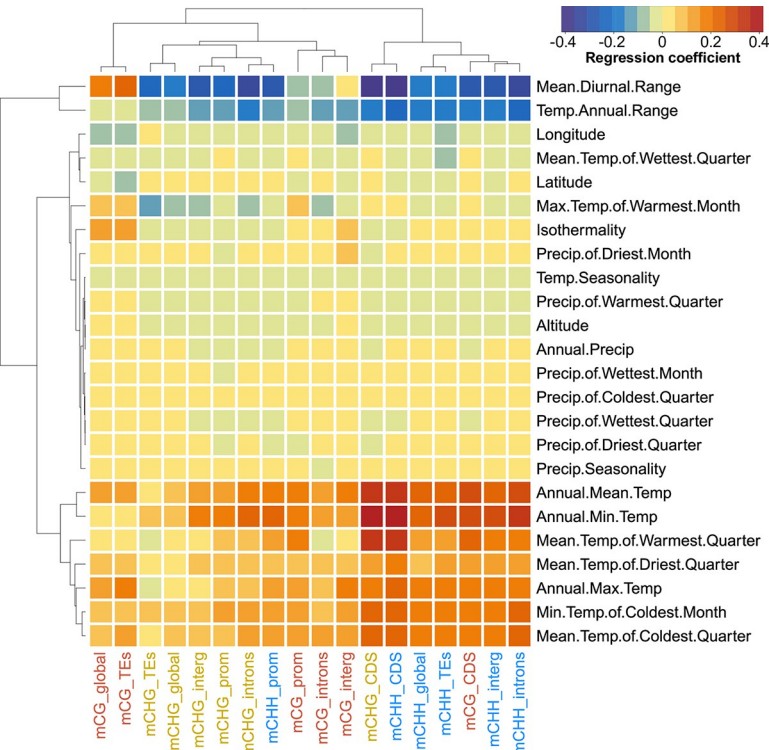

**Fig 4. Climate-methylation associations.** A Heatmap of the correlations between mean methylation and different climatic variables (Precip: precipitation; Temp: temperature), separately for different sequence contexts and genomic features (prom: promoter; interg: intergenic; TEs: Transposable Elements; CDS: coding sequences). Both rows and columns are clustered by their multivariate similarity in association patterns.

between climatic variables and the mean methylation in different sequence contexts and genomic features, correcting for population structure with the same IBS matrix used in the GWA analyses. We found that average methylation was positively correlated with several climate variables reflecting variation in mean temperatures, but negatively with variables related to temperature variability, such as the mean diurnal range and annual temperature range (Fig 4). Moreover, associations with temperature were more pronounced for minimum temperature variables than for maximum temperature variables. In other words, plants originating from colder origins or such with more fluctuating temperature environments had lower overall methylation. In contrast to the temperature variables, methylation was not associated with the precipitation variation of the population of origin, and there was also little association with latitude (Fig 4). The latter at first appears counterintuitive, because latitude is usually correlated with temperature, but in our case latitude is confounded with altitude–more southern samples were collected at higher elevations (S1 Table)–and therefore poorly correlated with temperature.

The described climate-methylation associations were generally stronger in CHG and CHH contexts, particularly for methylation that occurred in CDS (Fig 4). With the exception of mCG in CDS, which had climate associations similar to mCHH, other methylation variables clustered mostly by sequence context, with some similarity between CG and CHG. Finally, global and TEs mCG were the only types of methylation positively associated with temperature variability (Fig 4).

## DMR variance decomposition

Having established associations of methylation variation with genetic background and environment of origin, we sought to investigate the relative importance of these two drivers in our study system, and how this might vary between sequence contexts and genomic features. To address these questions, we analysed methylation variation at the level of DMRs. We identified around 44k DMRs in CG, 12k DMRs in CHG and 77k DMRs in CHH (see Methods for details on the DMR calling), and quantified their overlap with different genomic features. Most DMRs were located in TEs, and decreasing numbers in intergenic regions, promoters and genes (Fig 5B).

To quantify the degrees of genetic versus environmental determination, we then analysed three mixed models for each DMR that included either a distance matrix based on genetic variants in *cis*, on genetic variants in *trans*, or on multivariate climatic distances. Across all DMRs, genetic similarity based on *trans*-variants explained the largest proportions of methylation variance in all contexts (Fig 5A). Most variance was explained in CHG-DMRs, followed closely by CG-DMRs, but in CHH-DMRs the amounts of variance explained were generally much lower. Interestingly, the explanatory power of environmental variation relative to that of genetic variation gradually increased from the more stable mCG towards the less stable mCHG and mCHH (Fig 5A and 5C).

Although genetic variation in *trans* was on average the strongest predictor of methylation variance, there were large differences between individual DMRs, and we observed that

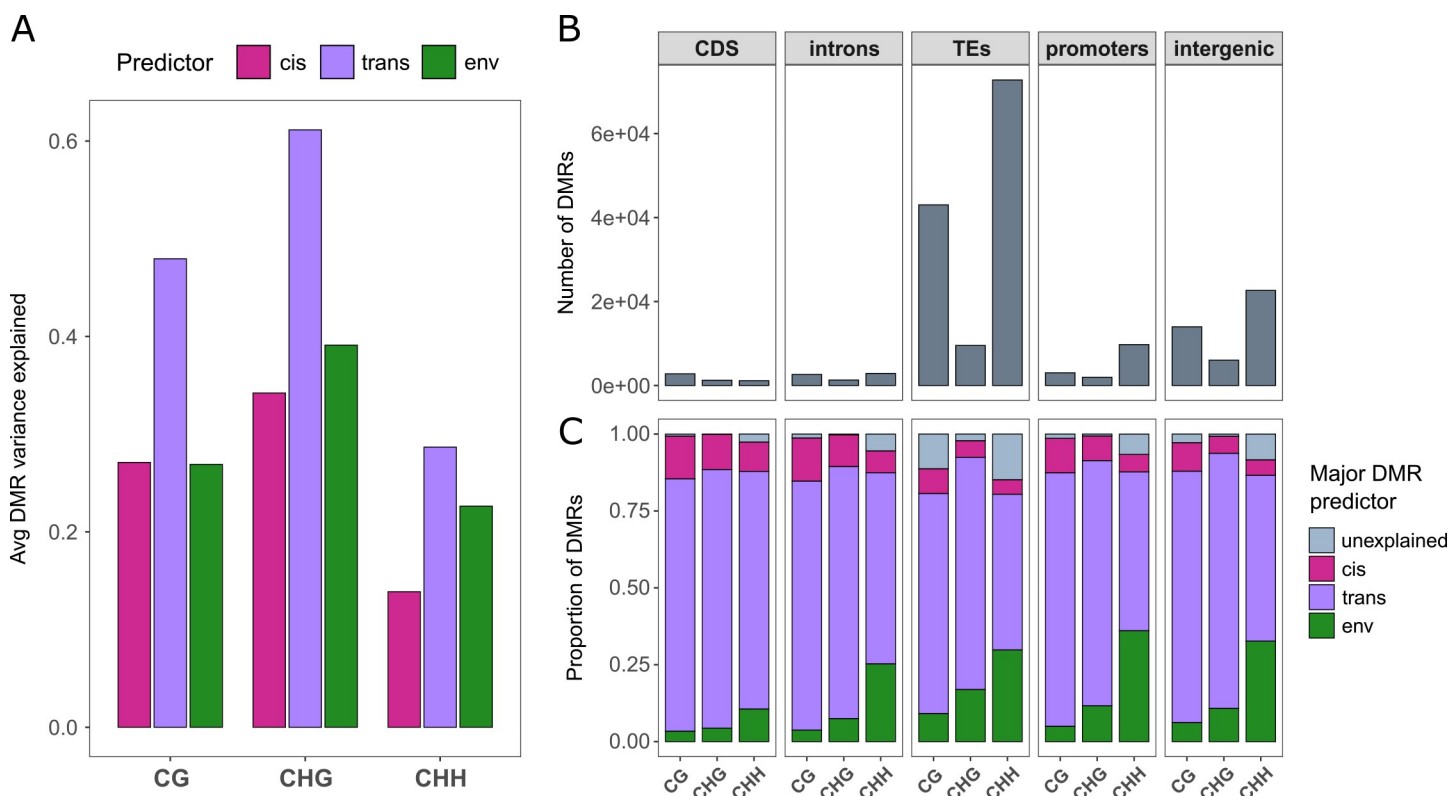

**Fig 5. Genetic versus environmental predictors of DMR variance.** (A) The variance in DMR weighted methylation explained by genetic similarity in *cis*, genetic similarity in *trans* and climatic similarity, averaged across all DMRs. (B) The number of DMRs identified in different genomic features and sequence contexts, and (C) the fractions of these individual DMRs where *cis*-variation, *trans*-variation or climatic variation are the major predictors. DMRs where none of the three predictors explained >10% of the variance are classified as "unexplained".

sometimes genetic variation in *cis* or climatic distance, too, could be the strongest predictor. To study this more systematically, we classified all DMRs based on their strongest predictor, and we found that the fraction of DMRs in which climate was a stronger predictor of methylation variance than any of the genetic distances increased from CG to CHG to CHH (Fig 5C). In CHH, 25–30% of all DMRs had climatic distance as their strongest predictor. To find out if *cis*-, *trans*- and climate-predicted DMRs were enriched close to genes responsible for different functions, we ran separate GO enrichment analyses for the genes neighbouring these three classes of DMRs. However, only for the *trans*-predicted DMRs we found significant enrichment of a few GO terms (S4 Fig), while there were none for the other two DMR classes.

## Discussion

Understanding natural epigenetic variation requires combining large-scale surveys of natural populations with high-resolution genomics and environmental data. Here, we studied European populations of *T. arvense* to assess how climate of origin and genetic background shaped their heritable DNA methylation variation. We found epigenetic population structure and confirmed the genomic patterns of methylation of the *T. arvense* genome [36] in a large natural collection. Most importantly, both genetic background and climate of origin were significantly associated with methylation variation, but their relative predictive power varied depending on DNA sequence context.

Our analysis of population structure detected two main clades, one composed of lines from all surveyed countries and a smaller one with almost exclusively lines from Sweden. A latitudinal gradient was also clear within the larger clade. The epigenetic population structure generally resembled the genetic one, with decreasing degrees of similarity from CG to CHG and to CHH sequence contexts (Fig 1B). These differences between contexts might reflect their different stability, caused by differences in the maintenance machineries [13,14] and possibly different proportions of genetic versus environmental control. Moreover, mCG shows stronger geographic patterns in genes and promoters than in TEs, possibly indicating a higher stability or selection for this kind of methylation (S1 Fig).

Across all lines, we calculated a global weighted methylation of 16.9%, which is high in the Brassicaceae family [20], particularly in comparison to *A. thaliana* (5.8%) [12]. The high global methylation is related to the high TE content of the *T. arvense* genome (~60%)[36], but also to a higher CHH methylation (12.3% across all lines) than it is known for most other angiosperms [20]. The levels of CG and CHG methylation (47.4% and 14.2% across all lines), in contrast, are more similar to other Brassicaceae [20]. As expected, we found that methylation was very unevenly distributed not only between sequence contexts, but also between genomic features, with high levels of methylation particularly in CG context, and in TEs and intergenic regions (Fig 2A). Gene body methylation was generally very low, with lines carrying on average ~93% of the CDS unmethylated in all contexts (S3 Table), and the results were similar, albeit much less extreme, for promoters (Fig 2B)[36]. When methylated, CDS were usually methylated in all contexts, showing TE-like patterns, while CG-only methylation, typical of many housekeeping genes in other species [20], was almost completely absent (S2 Fig). This uncommonly low gbM is present in other Brassicaceae [20], in particular in the close relative *Eutrema salsugineum* and might have evolved before speciation between Thlaspi and Eutrema [20,36]. Although the loss of *CHROMOMETHYLTRANSFERASE 3 (CMT3)* was previously associated to the loss of gene body methylation [51], this gene is expressed, although possibly mutated, in Thlaspi. If *CMT3* is indeed mutated in Thlaspi, the mutation is likely to affect all lines equally, since we found no variants neighbouring *CMT3* associated with variation in the

number of gbM genes. Instead a significant peak in Scaffold_7, pointing towards other genes, might explain this variation (S2C Fig). TE-like methylated genes, which are usually pseudo-genes in many species, were enriched for some constitutive functions and were about eight times more likely than average to overlap with TEs. This might indicate that the extensive TE expansion that occurred in the Thlaspi genome also affected some housekeeping genes, without compromising viability (S2 Fig). Overall these findings suggest that gene body methylation in *T. arvense* differs from most previously studied plant species [20].

To understand the genetic basis of methylation variation in *T. arvense*, we used GWA analyses, testing for associations between DNA sequence variation and average methylation levels in different sequence contexts and genomic features. With a strict Bonferroni correction, we did not detect any significant genetic variants, which probably resulted from a combination of our moderate number of only 207 sequenced lines, the nested sampling design, and the high number of tests (compared to *A. thaliana*) in a ~500 Gb genome. However, for some methylation phenotypes, we found strong enrichment of *a priori* candidates neighbouring genes known to play a role in DNA methylation from *A. thaliana* studies, and this indicates that many of our top peaks are likely to be true positives (Figs 3 and S3). Examples include the peaks detected next to the genes *AGO9*, *DRM3* and *LIG1*, which are all part of the DNA methylation machinery of *A. thaliana* [13,14], and which were also among our *a priori* candidates (S5 Table). In addition to these 'expected' candidates, we found several additional peaks next to genes that were indirectly linked to DNA methylation, with predicted functions such as histone acetylation, DNA repair and ubiquitination (S6 Table). The latter in particular is a post-translational modification which was previously shown to affect methylation in several ways [28–33]. These new candidate genes were not in our *a priori* list, which explains the drop of enrichment at high -log(p) in several GWA analyses (Figs 3B and S3). Our results show that while there appears to be partial overlap in the genetic control of DNA methylation between *T. arvense* and *A. thaliana*, there are also important differences. Some of our strongest candidates have not been associated with DNA methylation before, particularly not in natural populations. Functional characterization of these "new" candidates will be necessary to confirm our findings and understand the mechanisms of action of these genes.

Finally, some interesting associations warrant further exploration and could uncover functional differences with *A. thaliana* in the methylation machineries of different sequence contexts. For example we find a peak for mCG, next to a *DRM3* orthologue, involved in RdDM and non-CG methylation maintenance in Arabidopsis [46–48], and vice versa a peak for mCHH of promoters and TEs right next (3kb upstream) to an orthologue of the mCG maintenance methyltransferase *MET1*. On the contrary, the high similarity between mCHG and mCHH in regard to their genetic basis, as shown by the strong overlap of GWA results, seems to be a common feature in the plant kingdom [13,14].

Natural epigenetic variation was not only associated with genetic background in our study, but also with climate of origin. These correlations were generally much stronger than those with latitude or longitude, which supports the idea that the observed correlations reflect adaptive processes and not just the combination of epigenetic drift and isolation-by-distance. Specifically, we found average methylation to be positively correlated with mean temperature but negatively with temperature variability (Fig 4). Our field survey particularly captured the cold end of the distribution range of *T. arvense* (Mean Annual Temp. 6.5–11.1˚C). Previous studies showed that cold can induce DNA demethylation in plants [52–54] and that demethylation in turn can be associated with expression of cold-resistance genes and increased freezing tolerance [55,56]. The observed negative correlations between methylation and temperature might therefore reflect adaptation to cold and the fact that we captured the cold end of the distribution. This interpretation is further supported by the fact that correlations with minimum

temperatures were generally stronger than with bioclimatic variables capturing maximum temperature (Fig 4) and explains why a similar study found negative correlations between temperature and methylation in Arabidopsis accessions sampled on a range including many warmer locations [12]. The negative relationship between DNA methylation and temperature variability (Mean Diurnal Range and Temperature Annual Range) is more challenging to interpret, as there have so far been no experimental tests manipulating environmental variability in temperature. However, lower DNA methylation is often associated with lower genome stability [57,58], and it is conceivable that in fluctuating and thus less predictable environmental conditions, lower genome stability and higher transposon activity could be adaptive. Supporting this hypothesis, Arabidopsis *cmt2* mutants with slightly lower and more variable CHH methylation in TEs, were shown to be more common in regions with high temperature seasonality [59]. Finally, we did not find any association between DNA methylation and the precipitation of the population origins. However, this may largely be a result of our latitudinal sampling design, which maximized temperature but not precipitation variation. None of our sampling sites were particularly dry or particularly wet/oceanic (Annual Prec. 475–869 mm).

To better understand the predictive power of climate of origin versus genetic background, we finally analysed the variance in methylation levels of individual DMRs. We found that, across all DMRs, genetic variation in *trans* generally explained more DMR variation than climatic variation or genetic variation in *cis*. However, there was a trend from CG to CHG to CHH that the explanatory power of climate increased relative to that of genetic background (Fig 5A). In CHH, climate was the strongest predictor of methylation variation in over one quarter of the individual DMRs; in promoters this was true for even 35% of the DMRs (Fig 5B). These results further support the idea that methylation variation, particularly in CHG and CHH, is not only involved in plant responses to short-term stress [17] but also in longer-term environmental adaptation. Moreover, the observation that sometimes climate was the strongest predictor, indicates that at least part of the climate-methylation associations could be independent of DNA sequence variation [5]. Clearly, further work is needed to support these speculations, in particular high-resolution analyses that disentangle the genomic versus epigenomic basis of relevant phenotypes related to climatic tolerances. We attempted to get some hints of the functional basis of the observed genomic-methylation and climate-methylation relationships by analysing GO enrichment in the neighbouring genes of *trans-*, *cis-* and environmentally-associated DMRs, and we found some enrichment, mostly related to housekeeping functions, for *trans*-DMRs, but none for *cis-* and environmentally-associated DMRs (S4 Fig). However, the functional annotation had GO terms for only less than half of our candidate genes, so our GO enrichment analysis had rather limited power.

In summary, our study is the first large-scale investigation of DNA methylation variation in natural plant populations beyond the Arabidopsis model. We found that *T. arvense* natural DNA methylation variation is shaped by genetic and environmental factors, and that the relative contributions of the two drivers vary strongly between sequence contexts. Methylation variation in CG is generally the most similar to, and best predicted by, genetic variation. Moving to CHG and particularly CHH, the genetic determination decreases making environmental determination relatively higher. Our results thus indicate that DNA methylation could play a role in the large-scale environmental adaptation of *T. arvense*. Further experimental research, in particular dissecting adaptive phenotypes, is necessary to corroborate this hypothesis. There are currently efforts underway to develop *T. arvense* into a new biofuel and winter cover crop [37–41], and any insights into the genomic basis of climate and other environmental adaptation will be highly relevant to these efforts, particularly to deal with future climates.

## Materials and methods

### Sampling and plant growth

In July 2018, we collected *T. arvense* seeds from 36 natural populations in six European regions, spanning from southern France to central Sweden, and used them to conduct a common environment experiment in Tübingen, Germany. The experiment started at the end of August 2018 and lasted about two months. Upon sowing 207 lines in 9x9 cm pots filled with soil, we stratified them for 10 d at 4°C in the dark. We then transferred the seeds to a glasshouse and transplanted seedlings to individual pots upon germination. The glasshouse had a 15/9 h light/dark cycle (6 a.m. to 9 p.m.) with temperature and humidity conditions averaging 18°C and 30% at night and 22°C and 25% during the day. External conditions influenced these parameters, resembling natural growing conditions. 46 d after the end of the stratification period, we collected the 3rd or 4th true leaf and snap-froze it in liquid nitrogen.

### Library preparation and sequencing

Using the DNeasy Plant Mini Kit (Qiagen, Hilden, DE), we extracted DNA from disrupted leaf tissue obtained from the 3rd or 4th true leaf. For each sample, we sonicated (Covaris) 300 ng of genomic DNA to a mean fragment size of ~350 bp and used the resulting DNA for both genomic and bisulfite libraries. The NEBNext Ultra II DNA Library Prep Kit for Illumina (New England Biolabs) was used for library preparation and was combined with EZ-96 DNA Methylation-Gold MagPrep (ZYMO) for bisulfite libraries. Briefly, the procedure involved: i) end repair and 3' adenylation of sonicated DNA fragments, ii) NEBNext adaptor ligation and U excision, iii) size selection with AMPure XP Beads (Beckman Coulter, Brea, CA), iv) splitting DNA for bisulfite (2/3) and genomic (1/3) libraries, v) bisulfite treatment and cleanup of bisulfite libraries, vi) PCR enrichment and index ligation using Kapa HiFi Hot Start Uracil + Ready Mix (Agilent) for bisulfite libraries (14 cycles) and NEBNext Ultra II Q5 Master Mix for genomic libraries (4 cycles), vii) final size selection and cleanup. Finally, we sequenced paired-end for 150 cycles. Genomic libraries were sequenced on Illumina NovaSeq 6000 (Illumina, San Diego, CA), while bisulfite libraries were sequenced on HiSeq X Ten (Illumina, San Diego, CA).

### Variant calling, filtering and imputation

Base calling and demultiplexing of raw sequencing data were performed by Novogene using the standard Illumina pipeline. After quality and adaptor trimming using cutadapt v2.6 (M. Martin 2011), we aligned reads to the reference genome [36] with BWA-MEM v0.7.17 [60]. We then performed variant calling with GATK4 v4.1.8.1 [61,62] following the best practices for Germline short variant discovery (https://gatk.broadinstitute.org/hc/en-us/articles/360035535932-Germline-short-variant-discovery-SNPs-Indels-). Briefly, we marked duplicates with MarkDuplicatesSpark and ran HaplotypeCaller to obtain individual sample GVCF files. We combined individual GVCF files running GenomicsDBImport and GenotypeGVCFs successively and parallelizing by scaffold, obtaining single-scaffold multisample vcf files. We then re-joined these files with GatherVcfs. Upon assessment of quality parameters distributions, we removed low quality variants using VariantFiltration with different filtering parameters for SNPs (QD < 2.0 || SOR > 4.0 || FS > 60.0 || MQ < 20.0 || MQRankSum < -12.5 || ReadPosRankSum < -8.0) and other variants (QD < 2.0 || QUAL < 30.0 || FS > 200.0 || ReadPosRankSum < -20.0). Using vcftools v0.1.16 [63], we further filtered scaffolds with less than three variants and variants with multiple alleles or more than 10% missing values. Prior to imputation, we only applied a mild Minor Allele Frequency (MAF) > 0.01 filtering not to

reduce imputation accuracy [64]. Imputation with BEAGLE 5.1 [65] recovered the few missing genotype calls left, outputting a complete multisample vcf file.

## Methylation analysis

The EpiDiverse WGBS pipeline is specifically designed for bisulfite reads mapping and methylation calling in non-model species (https://github.com/EpiDiverse/wgbs) [66]. We used it to perform quality control (FastQC), base quality and adaptor trimming (cutadapt), bisulfite aware mapping (erne-bs5), duplicates detection (Picard MarkDuplicates), alignment statistics and methylation calling (Methyldackel). In the mapping step, we only retained uniquely-mapping reads longer than 30bp. The pipeline outputs context-specific (CG, CHG and CHH) individual-sample bedGraph files, which we filtered for coverage > 3 and combined in multisample unionbed files with methylated/total read counts for every position and sample (we used custom scripts and bedtools [67]). We retained all cytosines with coverage > 3 in at least 75% of the lines and used this dataset for all subsequent analyses.

For describing general patterns of methylation, we calculated weighted methylation as the fraction between all methylated and all total read counts at every cytosine included in the calculation [43]. In this way we also calculated the bisulfite non-conversion rates, including all cytosines with coverage > 10 [2] in two regions of Scaffold_364 (51–60.5 KB and 95–110 KB), selected for high similarity to chloroplast DNA and confidently unmethylated. For analyses of variation between lines (GWA and correlation with climate) we used mean methylation, which is obtained by calculating the methylation of each position first (methylated/total read count) and then averaging all positions included in the calculation [43]. Weighted methylation corrects for coverage, but is highly influenced by structural and copy number variants, which are likely abundant in a species with such a high TE content [36]. As we were interested in true variation of methylation levels, mean methylation was more suited for comparing methylation of whole genomic features between lines.

To extract the mean and weighted methylation of genomic features, we intersected (bedtools) [67] unionbed files with genomic features (genes, CDS, introns, TEs, promoters and intergenic regions) and averaged methylation of all intersected cytosines. For introns, we only included regions annotated as intronic on both strands. We also extracted weighted methylation of individual CDS, promoters and TEs across all samples and plotted their distributions. We then used this information to calculate the mean methylation of genes, excluding lowly methylated ones (average mC < 5% across all lines) and used it for GWA. For PCA, we used the R [68] function prcomp(). Genome wide PCAs were only based on positions without missing values as these were already a large amount (always > 1 million). Instead when restricting to genomic features we allowed for 2% NAs and imputed these with the "missMDA" R package [69] to include a larger amount of positions (always > 0.8 million). Nevertheless comparison of PCA plots with and without imputation gave very similar results.

## Gene Body Methylation classification

To test whether genes were methylated in their CDS, in any of the sequence contexts, we adopted a method from previous authors [20,45]. First we used a binomial test to determine, for each cytosine in CDS, whether it had significantly higher methylation than expected from bisulfite non-conversion rates (P < 0.01). We then computed the fraction of methylated cytosines in all CDS and lines, separately for each sequence context. Finally we tested if the fraction of methylated cytosines of each individual CDS, was higher than the average of all CDS, with a one-sided binomial test. In other words, we tested whether a specific CDS had a higher density of methylated positions than all CDS on average. Upon correcting for multiple testing with the p.adjust() R function [68], we considered "methylated" CDS with FDR<0.05. We restricted the analysis to genes with at least

10 covered cytosines (coverage > 3) in each sequence context, for at least 90% of the lines. If a CDS had less than 6 cytosines covered in a line, we coded it as a missing value. Such analysis revealed the methylation status of 22703 genes in each line and sequence context. We defined as "gbM", genes with mCG FDR < 0.05, and mCHG and mCHH FDR > 0.05. We defined as "teM", genes with mCHG or mCHH FDR < 0.05 [12]. For GO enrichment analysis we used genes consistently methylated, i.e. methylated in at least 70% of the lines.

## Population genetic and GWA analysis

For basic genetic population structure analysis, including PCA plots and generation of the IBS matrix, we applied a mild MAF filtering (MAF>0.01) and performed variants pruning with PLINK v1.90b6.12 [70], using a window of 50 variants, sliding by five and a maximum LD of 0.8. Upon this filtering, we also used PLINK to generate the IBS matrix used in several analyses to correct for population structure or for DMRs variance decomposition. For PCA, we used the R [68] function prcomp().

We ran GWA analyses for multiple phenotypes using a custom script based on the R package "rrBLUP" [71], which allows to run mixed models correcting for population structure with the above-mentioned IBS matrix. We used biallelic variants and applied a MAF > 0.04 cutoff. For Manhattan and QQplots we used the "qqman" package [72], calculating the genome-wide significance threshold according to Sobota et al. (2015) [49]. We ran GWA analyses using each average methylation context (CG, CHG and CHH) feature (global, CDS, introns, TEs, promoters and intergenic regions) combination as phenotype. For genes we also calculated mean methylation of methylated genes, excluding lowly methylated ones (average methylation > 5% across all lines), ending up with a total of 24 methylation phenotypes (S4 Table). Since a few samples had higher than usual non-conversion rates (S2 Table), leading to an overestimation of their average methylation, we calculated, for each individual sample, the surplus non-conversion rate from a baseline of 0.6%, and subtracted it from the mean methylation values. The baseline of 0.6% allowed us to correct only the ~20% of samples with highest non-conversion rates. Occasionally, we observed a positive correlation between mean methylation and coverage across lines, probably due to library preparation bias. In these cases we fit a linear model to the data using the logarithm of coverage (from bam files), as this gave the best fit in all cases, and used the residuals for GWA analysis. Finally, we applied Inverse Normal Transformation to mean methylation phenotypes that deviated strongly from normality. A list of all methylation phenotypes used and corrections and transformations applied, can be found in S4 Table.

With the double aim of validating GWA results and comparing with previous *A. thaliana* studies, we performed enrichment of variants neighbouring *a priori* candidate genes, according to the method established by Atwell et al. (2010) [50]. We made a few additions to the methylation candidate gene list used by Kawakatsu et al. (2016) [12], kindly provided by the authors, extracted all *T. arvense* orthologues that we could retrieve from orthofinder [73] analysis and used them for our *a priori* candidate genes list (S5 Table). Briefly, we attributed "*a priori* candidate*" status to all variants within 20kb from genes in the list and calculated enrichment for increasing -log(p) thresholds as the ratio between Observed frequency (sign. *a priori* candidates/sign. variants) and Background frequency (total *a priori* candidates/total variants). Using the same formula adopted by Atwell et al. (2010) [50], we additionally calculated an upper bound for the FDR among candidates.

## Climate-methylation correlations

To obtain bioclimatic variables for the 25 years predating the experiment, we downloaded temperature and precipitation variables from the "E-OBS daily gridded meteorological data for

Europe" database (v21.0), freely available on the Copernicus website [44]. All downstream analyses were conducted in R [68]. We extracted data for our population locations with the "ncdf4" package [74], calculated monthly averages and extracted bioclimatic variables with "dismo" [75]. Finally, we averaged bioclimatic variables from 1994 to 2018, the year of collection (S7 Table). To test for climatic patterns in methylation, we ran mixed models for all mean methylation variables (the same as we used for GWA) and bioclimatic variables combinations, using the relmatLmer() function from the R package "lme4qtl" [76] and correcting for population structure using the same IBS matrix used for GWA analysis.

## DMR calling

The EpiDiverse toolkit [66] includes a DMR pipeline based on metilene [77], which calls DMRs between all possible pairwise comparisons between user-defined groups. We used this tool to call DMRs using the 36 populations as groups, a minimum coverage of five (cov > 4) and default values for all other parameters. We complemented the pipeline with a custom downstream workflow to obtain DMRs for the whole collection from comparison-specific DMRs. Briefly, since the pipeline output had an enrichment of short and close DMRs (particularly in CHH), we joined all comparison-specific DMRs that were closer than 146bp and had the same directionality (higher methylation in the same group). 146bp was chosen for consistency with the pipeline fragmentation parameter. We then merged DMRs from all pairwise comparisons (bedtools) [67] in a unique file and re-extracted weighted methylation of the resulting regions from all samples. Finally, we filtered DMRs with a minimum methylation difference of 20% (CG) or 15% (CHG and CHH) in at least 5% of the samples. This ensured to select DMRs with variability at the level of the whole collection.

## DMR variance decomposition

To quantify the variance in methylation explained by *cis*-variants, *trans*-variants, and by the environment, we ran three mixed models for each individual DMR using the marker_h2() function from the R package "heritability" [78]. Each model had one random factor matrix, capturing one of the three predictors. For *cis* we used an IBS matrix generated with PLINK v1.90b6.12 [70] from variants within 50kb from the DMR middle point. For *trans* we used the same IBS matrix used for all other analyses, described in the previous chapter. For the environment we calculated the Euclidean distance between locations, based on all Bioclimatic Variables averaged over 25 years before the sampling (1994–2018), and further reversed and normalized the matrix to obtain a similarity matrix in a 0 to 1 range. To summarize the results we: i) averaged *cis*, *trans* and environment explained variance across all DMRs and ii) classified each DMR based on the mayor predictor.

## Supporting information

**S1 Fig. PCA plots of all 207 lines.** (A) Complement to Fig 1B with latitude-coloured PCA plots for the missing PC. (B) latitude-coloured PCA plots based on methylation of specific genomic features (genes, TEs and promoters).
(PDF)

**S2 Fig. Genes methylated in each context, GO enrichment analysis and GWA.** (A) Venn diagram of the number of genes methylated in each context in at least 70% of the lines, which were also used for the GO enrichment. Genes methylated only in CG are labelled as "gbM", genes methylated in either CHG or CHH as "TE-like" [12]. (B) GO enrichment analysis of methylated genes corresponding to (A). Only significant results for GO terms with minimum

gene count of four are reported. GO categories are: Biological Process (BP), Cellular Component (CC) and Molecular Function (MF). (C) GWA for number of gbM genes, including Manhattan plot, enrichment of *a-priori* candidates and qqplot.
(PDF)

**S3 Fig. Complete methylation GWA results.** Manhattan plots, enrichment of *a priori* candidate variants and QQplots for all mean methylation phenotypes. more5met: mean methylation of genes with methylation > 5% across all lines. The genome-wide significance (horizontal red lines), was calculated based on unlinked variants as in Sobota et al. (2015) [49], the suggestive-line (blue) corresponds to–log(p) = 5. Top variants are labelled with the neighbouring genes potentially affecting methylation.
(PDF)

**S4 Fig. GO enrichment analysis of genes neighbouring trans-DMRs.** Genes neighbouring (2kb max) *cis*, *trans* and *env*-DMRs were used for individual GO term enrichment analysis, but only the *trans*-DMRs gene set was enriched for any significant term.
(PDF)

**S1 Table. Geographic locations of all *T. arvense* populations.** Geographic coordinates, elevation and size of all populations.
(PDF)

**S2 Table. Mapping statistics.** Number of deduplicated mapped reads, average coverage and non-conversion rates calculated from chloroplast DNA. WGS: Whole Genome Sequencing; WGBS: Whole Genome Bisulfite Sequencing.
(CSV)

**S3 Table. Number of genes methylated in each line.** Numbers and fractions of genes per line methylated in each sequence context, in CG only (gbM) and in either CHG or CHH (TEm) [12].
(XLSX)

**S4 Table. List of all mean methylation variables used for GWA and climate correlations.** Coverage correction indicates that, prior to GWA, residuals were extracted from a linear model with log(coverage) as predictor. INT indicates Inverse Normal Transformation. more5-met: Mean methylation of genes with methylation > 5% across all lines.
(PDF)

**S5 Table. List of *Thlaspi arvense a priori* candidate genes.** *T. arvense* genes and the respective *A. thaliana* orthologues with known roles in methylation. We used this list for the enrichment of a priori candidate variants performed upon GWA.
(CSV)

**S6 Table. GWA candidate genes.** List of all genes located within 15kb from variants significant to -log(p) > 5, including methylation phenotypes where the association was found, *a priori* candidate status and relevant functional putative roles. Genes with predicted function possibly affecting methylation are highlighted in bold.
(XLSX)

**S7 Table. Bioclimatic variables.** Bioclimatic variables used in this study, obtained from monthly averages extracted from the Copernicus programme website [44] and averaged for 1993–2018.
(CSV)

## Acknowledgments

We thank the entire EpiDiverse network for its amazing support and discussions, in particular Adrián Contreras-Garrido for providing orthofinder results and discussing analysis, and Bárbara Díez Rodríguez and Iris Sammarco for really useful suggestions. We thank Detlef Weigel for his input on data analysis, and Magnus Nordborg and Eriko Sasaki for their feedback on GWA analysis and for sharing their list of candidate genes. Finally, we thank Anupoma Troyee and Valentina Vaglia for helping with sampling, Sabine Silberhorn, Christiane Karasch-Wittmann, Eva Schloter, Julia Rafalski and Elodie Kugler for the greenhouse experiment, and Katharina Jandrasits for help with library preparation. For computing, we acknowledge Prof. Peter Stadler at the University of Leipzig and David Langenberger from ecSeq, for hosting the EpiDiverse servers. We also acknowledge the High Performance and Cloud Computing Group at the Zentrum für Datenverarbeitung of the University of Tübingen for managing the BinAC server.

## Author Contributions

**Conceptualization:** J. F. Scheepens, Claude Becker, Oliver Bossdorf.

**Data curation:** Dario Galanti, Adam Nunn, Isaac Rodríguez-Arévalo.

**Formal analysis:** Dario Galanti.

**Funding acquisition:** Claude Becker, Oliver Bossdorf.

**Investigation:** Dario Galanti, Daniela Ramos-Cruz.

**Methodology:** Dario Galanti, Oliver Bossdorf.

**Project administration:** Oliver Bossdorf.

**Resources:** Daniela Ramos-Cruz, Claude Becker.

**Software:** Dario Galanti, Adam Nunn.

**Supervision:** J. F. Scheepens, Claude Becker, Oliver Bossdorf.

**Validation:** Dario Galanti.

**Visualization:** Dario Galanti.

**Writing – original draft:** Dario Galanti, Oliver Bossdorf.

**Writing – review & editing:** Dario Galanti, Daniela Ramos-Cruz, Adam Nunn, Isaac Rodríguez-Arévalo, J. F. Scheepens, Claude Becker, Oliver Bossdorf.

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
