## [Decision Letter · Decision Letter 0]

28 Apr 2022

Dear Dr Galanti,

Thank you very much for submitting your Research Article entitled 'Genetic and environmental drivers of large-scale epigenetic variation in Thlaspi arvense' to PLOS Genetics.

The manuscript was fully evaluated at the editorial level and by independent peer reviewers. The reviewers appreciated the attention to an important problem, but raised some substantial concerns about the current manuscript. Based on the reviews, we will not be able to accept this version of the manuscript, but we would be willing to review a much-revised version. We cannot, of course, promise publication at that time. 

The reviewers had two different potential major issues that will both need to be addressed.  The third reviewer expressed concern about the approaches being used to classify per gene methylation types.  It will be important to provide detail on whether intron/UTR sequences are being retained and the specific details of this approach.  On a side note, working in maize we have noticed a large number (thousands) of genes with intronic TEs and the choice of whether to include introns or not when classifying per gene methylation has a major influence.  The first and second reviewers did not have specific concerns about the science but questioned the novelty of this work.  It will be important to clearly describe and emphasize the novelty of the results to make it clear that this is not simply showing the same results/patterns in a second species (compared to A. thaliana).  In addition to this major concerns the reviewers also made a number of helpful specific suggestions that could be incorporated to improve the manuscript. 

If you decide to revise the manuscript for further consideration at PLOS Genetics, please aim to resubmit within the next 60 days, unless it will take extra time to address the concerns of the reviewers, in which case we would appreciate an expected resubmission date by email to plosgenetics@plos.org.

[LINK]

We are sorry that we cannot be more positive about your manuscript at this stage. Please do not hesitate to contact us if you have any concerns or questions.

Yours sincerely,

Nathan M. Springer

Associate Editor

PLOS Genetics

Wendy Bickmore

Section Editor: Epigenetics

PLOS Genetics

Reviewer's Responses to Questions

**Comments to the Authors:**

Reviewer #1: Beyond Arabidopsis, profiling of natural diversity in the patterns of DNA methylation in large populations is scarce, and especially profiling using whole genome bisulfite sequencing. Thus, this manuscript is quite novel in that respect; the authors profile genome-wide patterns of DNA methylation in 207 pennycress lines, sampled from 36 natural populations. Both genetic background and climate of origin were found to be significantly associated with methylation variation.

This manuscript was a pleasure to read, thoughtful and polished.

Overall, the notable findings were quite similar to prior work in Arabidopsis. SNPs were found associated with variation in average methylation for different genomic features; and candidate genes were identified including known and novel candidate players. Average methylation levels show an association with climate of origin, colder climates are associated with reduced methylation, particularly CHH. It is interesting too that CG/CHG at TEs is negatively associated with the max temp/month climatic variable. Many DMRs also showed an association with both climate of origin and SNPs, with trans-SNPs being the strongest predictor. It would be great to have paired expression data, but that would be a substantial undertaking (not suggesting it).

These are definitely notable findings and great to see replicated beyond Arabidopsis. I guess one thing I wanted to discover is a strong argument to support why we do indeed need to do these studies in plants with larger, more TE rich genomes! I think we do, but do these results suggest that Arabidopsis really is a good enough model after all? Perhaps some more compare/contrast with the major conclusions from Arabidopsis work could be included in the discussion?

Otherwise, I have just a couple of minor suggestions that should be addressed:

DRM3:

DRM3 is identified as a candidate - it has been reported previously that members of the DRM3 clade in plants lack conserved residues important for cytosine methyltransferase activity (https://doi.org/10.1371/journal.pgen.1001182 and doi: 10.1186/1756-0500-7-721); and hence while DRM3 affects methylation levels, it is likely a catalytically inactive paralog that promotes DRM2 activity. Based on the genome sequence, would the pennycress DRM3 be predicted to be active on inactive?

SNP-DMR associations:

Genetic variance in trans explained the largest proportion of DMRs (line 295), wondering if this is associated with many different SNPs or are there a few large effect SNPs (‘hotspots’) associated with many DMRs (eg AGO9, DRM3 or even CMT?)? Perhaps a chromosomal dot plot for associations of DMR vs SNP along the chromosomes could help visualise this (eg see Fig 3 in Dubin et al 2015 https://doi.org/10.7554/eLife.05255.007)? If there are some large effect SNPs/genes, should these be added to the model, especially if temperature also affects average methylation – could there be an environment x major-effect-SNP interaction?

Minor:

- it could be useful to label the major geographic collection locations in Fig 1a for non-European readers?

- line 148 it would be good to also include a table (or the state the average) conversion rate and mapping rate.

- line 257 are the bioclimatic data for each location available somewhere? I couldn’t see them but these would be required to replicate the findings; can a supp table be added.

- line 288 – the DMR list and/or co-ordinates might also be provided eg on Zendo.

- line 467 please provide a specific reference or link to the ‘reference genome’; what was the reference used?

Reviewer #2: The manuscript by Galanti and colleagues describes analysis of factors underlying methylation variation in 207 lines of pennycress. Genetic variation is the main driver of global methylation variation, however climate of origin also has a small impact. The results largely recapitulate what has been demonstrated previously in Arabidopsis, making the manuscript of narrower interest. Generally, the analysis is sound and the conclusions justified, however I have a few suggestions for improvement.

Major points:

Lines 79-81: The statement that CHH methylation is maintained by persistent de novo methylation by RdDM is only true in euchromatin. Throughout heterochromatin, CHH methylation is maintained by CMT2 in a mechanism similar to maintenance of CHG methylation. It is preferable to separate these two types of methylation during analysis. For example, changes in CMT2 alleles impact CHH methylation in heterochromatin but not in euchromatin (see Sasaki et al 2019 PLoS Genetics). I recognize that heterochromatin might not be clearly defined in pennycress, but perhaps proximity to genes could be used as a proxy, or ChIP-seq analysis of histone modifications could be completed in a reference line.

The abstract mentions 207 lines, but really it is 36 populations that were studied. The smaller number of independent populations might explain the limited statistical power of the GWAS.

Clearly genetic variation is the main driver of global methylation variation. However, the manuscript really favors the hypothesis that climate of origin is an important factor. While Figure 5C is compelling that CHH methylation is more likely to be explained by environmental factors than other contexts of methylation, this is still minor compared to the effects of genetic variation. The manuscript would be improved by a more balanced approach to the results.

It is surprising that the manuscript does not include discussion of Shen et al, PLoS Genet 2014, which demonstrated that alleles of CMT2 have been selected in variable climates (in Arabidopsis).

There is no mention of how bisulfite conversion was assessed in these libraries. Given the higher-than-expected CHH methylation values, analysis of chloroplast sequences or lambda phase spike-ins is essential.

Minor points:

I was unable to find an entry for PRJEB50950 in the ENA Sequence read archive.

Line 110: Methyltransferase 1 is commonly abbreviated as MET1, or to be consistent with non-plant literature, DNMT1, not DMT1.

Figure 3D says NRPB1L rather than NRPB10L

Reviewer #3: Galanti et al. provides a comparative analysis and GWAS of methylation across natural populations of Thlaspi arvense. Based on their analysis, genetic factors are the major driver of DNA methylation variation It also appears that this is a species that has potentially lost most of its gbM, however, I have major concerns about the robustness of the DNA methylation analysis that need to be addressed. It seems unlikely to me that these corrections could be completed in a short period.

1. I may have missed it, but I did not see any mapping statistics rates for any of the samples. For whole-genome bisulfite data, conversion or non-conversion rates are necessary. This is essential to evaluate the quality of the data.

2. It is not clear how the authors are calculating their methylation levels. Multiple approaches are used in the literature (see discussion in Schultz et al. https://www.cell.com/trends/genetics/fulltext/S0168-9525(12)00171-0). Some methods of calculation are strongly affected by factors such as read coverage. The preferred method is the “weighted methylation level” and allows for a more direct comparison between species. Also, by limiting their analysis to those cytosines with coverage in all samples, the author’s may also be missing the direct effects of genetic variation on methylation levels.

3. Line 185-196, Fig S2. I have a major concern regarding these results. It is highly unusual for genes with TE-like methylation (TeM) in greater abundance than CG-only gene-body methylation (gbM). While there are exceptions, such as Eutrema salsugineum, which have lost gbM, I suspect that the author’s results are due to inclusion of intronic/UTR methylation. This is supported by the GO-term analysis, which shows enrichment of houskeeping genes in TeM methylated patterns. In species like Eutrema, these genes are unM, while typically they are gbM in other species. From a straight-forward reading of the author’s methods, they are using DNA methylation within gene-bodies, including UTRs, CDS, and introns. Plant genes frequently have intronic TEs which contain extensive non-CG methylation. This is true even on CG-only gene-body methylated genes (gbM). If intronic cytosines are not excluded from such an analysis, this can lead to misinterpretation and misclassification of gbM genes as TE-like non-CG methylation. UTRs can also be problematic, as inaccurate UTR annotation in many species can also lead to over-representation of non-CG methylation in genes (see Niederhuth et al. https://genomebiology.biomedcentral.com/articles/10.1186/s13059-016-1059-0). I would also recommend a more rigorous approach for classification, such as those based on a binomial test of methylation in each gene, originally described by Takuno & Gaut (see https://academic.oup.com/mbe/article/29/1/219/1749122?login=true).

4. If the number of gbM genes is indeed so low, then the authors should check for the presence/absence CMT3 and other mutations.

**Have all data underlying the figures and results presented in the manuscript been provided?**

Reviewer #1: **No: **line 257 - are the bioclimatic data for each location available somewhere?

Reviewer #2: **No: **Methylation data should be available via the ENA sequence read archive, but I couldn't find them.

Reviewer #3: Yes

PLOS authors have the option to publish the peer review history of their article (what does this mean?). If published, this will include your full peer review and any attached files.

Reviewer #1: No

Reviewer #2: No

Reviewer #3: No

---

## [Decision Letter · Decision Letter 1]

16 Aug 2022

Dear Dr Galanti,

Thank you very much for submitting your Research Article entitled 'Genetic and environmental drivers of large-scale epigenetic variation in Thlaspi arvense' to PLOS Genetics.

The revised manuscript was evaluated by the peer reviewers that evaluated the initial submission and appreciated the revisions as these satisfied most of the concerns. Reviewers 2 and 3 raised one important concern that needs to be addressed prior to acceptance of the manuscript.  Both reviewers note that a subset of the samples have an elevated non-conversion rate.  We therefore ask you to modify the manuscript according to the review recommendations. Your revisions should address the specific points made by each reviewer.

[LINK]

Yours sincerely,

Nathan M. Springer

Academic Editor

PLOS Genetics

Wendy Bickmore

Section Editor

PLOS Genetics

Reviewer's Responses to Questions

**Comments to the Authors:**

Reviewer #1: My concerns and suggestions have all been addressed, thank you.

Reviewer #2: The authors have addressed each of the points I raised, however now that I can see the bisulfite conversion rates (supplemental table 2), I am concerned that almost 10% of the samples have quite high non-conversion rates (>1%). I would have thrown out any sample that converted this poorly because it inflates CHH signals by a substantial margin. The authors argue that even considering the poor non-conversion rates, CHH methylation is high for Brassicaceae. I have no quibbles with that conclusion, given most of the samples look good. However, I do not have enough experience with population-level methylation analysis to know whether such a distortion might impact the findings of this paper. I am concerned because the poorly converted samples are not randomly distributed in the set. For examples, TA_SE_01 has an average non-conversion of 1.25%, with 5 of 6 libraries poorly converted, while TA_DE_06 has an average non-conversion of 0.17%. Perhaps the editor or another reviewer has more experience and can weigh in on this concern. Alternatively, perhaps the authors can confirm their findings with the poorer libraries removed from the analysis.

Reviewer #3: I thank the authors for addressing my comments and the extensive work they have done.

I only have one comment on this latest version. In looking at the non-conversion rates in table S2, it looks like the lines with a high non-conversion (>1%) cluster together in the table (rows 112, 122-129). Are these lines are related or collected from a similar region? Since the authors used mapping to the chloroplast genome to calculate the non-conversion rate, the higher non-conversion rate in these lines could be indicative of nuclear insertion of chloroplast sequences. Such sequences are usually methylated. If not present in the genome assembly, these reads could map to the chloroplast and result in the perception of a higher non-conversion rate than there actually is. I do not in think that this will affect the results of the paper. The non-conversion rates are still low enough that they are unlikely to have much impact and if I am right that this is maybe an artifact of a nuclear chloroplast insertion, then the actual non-conversion is much lower. However, the authors should probably check whether these samples show any odd clustering that might affect their results.

**Have all data underlying the figures and results presented in the manuscript been provided?**

Reviewer #1: Yes

Reviewer #2: Yes

Reviewer #3: Yes

PLOS authors have the option to publish the peer review history of their article (what does this mean?). If published, this will include your full peer review and any attached files.

Reviewer #1: No

Reviewer #2: No

Reviewer #3: No

---

## [Editor Report · Decision Letter 2]

28 Sep 2022

Dear Dr Galanti,

We are pleased to inform you that your manuscript entitled "Genetic and environmental drivers of large-scale epigenetic variation in Thlaspi arvense" has been editorially accepted for publication in PLOS Genetics. Congratulations!  Thank you for the clear response to the potential issue of lower conversion rates for some genotypes.  I understand how it can be difficult with some data with lower quality and appreciate that you took the time to assess how this might influence the results and provide a method to reduce the impact of this issue.

Yours sincerely,

Nathan M. Springer

Academic Editor

PLOS Genetics

Wendy Bickmore

Section Editor

PLOS Genetics

Comments from the reviewers (if applicable):

**Data Deposition**

http://datadryad.org/submit?journalID=pgenetics&manu=PGENETICS-D-22-00367R2

**Press Queries**

---

## [Editor Report · Acceptance letter]

7 Oct 2022

PGENETICS-D-22-00367R2 

Genetic and environmental drivers of large-scale epigenetic variation in *Thlaspi arvense*

Dear Dr Galanti, 

We are pleased to inform you that your manuscript entitled "Genetic and environmental drivers of large-scale epigenetic variation in *Thlaspi arvense*" has been formally accepted for publication in PLOS Genetics! Your manuscript is now with our production department and you will be notified of the publication date in due course.

With kind regards,

Agnes Pap

PLOS Genetics

On behalf of:
